# A novel TRPV5/6-like channel from a scleractinian coral

**Angélica Méndez-Reséndiz**[1], **Gisela E. Rangel-Yescas**[1], **Miguel Benítez-Angeles**[2], **Tamara Rosenbaum**[2], **León D. Islas**[1]*

**1** Departamento de Fisiología, Facultad de Medicina, UNAM, México City, México, **2** Departamento de Neurociencia Cognitiva, Instituto de Fisiología Celular, UNAM, México City, México

* leon.islas@gmail.com

## Abstract

The calcium regulation mechanisms that underlie skeleton formation in stony corals are poorly understood. In epithelial tissues from vertebrates, transient receptor potential vanilloids 5 and 6 (TRPV5 and TRPV6), members of the TRP channel superfamily, play a significant role in transepithelial $Ca^{2+}$ transport. Particularly, TRPV5 is a constitutively active channel with a primary function in the $Ca^{2+}$ reabsorption mechanism of renal epithelium. It is characterized by a marked inward rectification and a high $Ca^{2+}$ permeability at physiological resting membrane potentials. Here, we report the cloning and characterization of a gene that encodes a protein homologous to the inward-rectifier cation channel TRPV5 in the reef-building coral *Pocillopora damicornis*. We assessed its biophysical properties and found that this channel displays inwardly rectifying $Na^+$ currents in the absence of divalent cations and can permeate $Ca^{2+}$, similar to the human TRPV5 channel. When compared to the human TRPV5, the specific blocker of this channel, miconazole, decreased the currents in a dose-dependent manner but did not affect the coral TRPV5/6-like-mediated currents. Interestingly, a monoterpene that has been shown to produce bleaching in corals, is also a blocker of the TRPV5/6-like channel. Altogether, our findings identify for the first time a novel TRPV5/6-like channel in scleractinian corals, whose potential physiological functions may include $Ca^{2+}$ transport to support the calcification mechanism.

## Introduction

The main contributors to the structural framework in reef ecosystems are corals. The structure of these ecosystems is maintained by the efficient production of calcium carbonate ($CaCO_3$), with scleractinian corals supporting the main portion of total coral reef $CaCO_3$ formation [1]. Scleractinian corals are marine, calcifying invertebrates belonging to the phylum Cnidaria, also known as stony corals [2]. Stony corals are colonial organisms composed of numerous polyps. Each polyp consists of two tissue layers: an oral layer in which some epithelial cells form symbiosis with dinoflagellates

**Data availability statement:** The data underlying the results are freely available and presented within the manuscript itself.

**Funding:** Dirección General de Asuntos del Personal Académico, Programa de Apoyo a Proyectos de Investigación e Innovación Tecnológica (DGAPA-PAPIIT) Grant No. IN201824. Fronteras de la Ciencia grant No. FC-513 from Consejo Nacional de Humanidades, Ciencia y Tecnología The funders had no role in study design, data collection and analysis, decision to publish, or preparation of the manuscript.

**Competing interests:** The authors have declared that no competing interests exist.

of the genus *Symbiodinium* and an aboral tissue layer in which the epithelium participates in the calcification process [3]

The endosymbiotic relation transfers organic matter from photosynthesis to the coral to meet most primary carbon needs [4,5]. The symbiosis can break down under stressful, high seawater temperature conditions, leading to symbiont expulsion and coral bleaching. However, the mechanism is not well understood [6]

The calcification reactions occur in the extracellular calcifying medium (ECM) adjacent to the aboral tissue, a specialized compartment containing the calcifying fluid where the mineral substrate precipitates and requires a constant supply of calcium ($Ca^{2+}$) ions to precipitate $CaCO_3$ (Allemand et al. 2011) and remotion of $H^+$ ions, rendering corals sensitive to ocean acidification [7,8].

The calicodermis is directly in contact with the ECM and the skeleton and is the tissue layer with the most direct role in calcification. These cells modify the composition of the ECM by regulating the supply of ions either actively through a transcellular route against their electrochemical gradient or passively via a paracellular pathway following the ion gradient [3,9]. It is noteworthy that in the case of $Ca^{2+}$, the pathway involves a transcellular route because the concentrations in the ECM range from 13 mM and are higher than seawater levels, estimated at 11 mM [10].

Based on existing data, it has been suggested that $Ca^{2+}$ enters the calicodermis via L-type $Ca^{2+}$ channels [11] and is released to the ECM through a $Ca^{2+}$- ATPase, which exchange protons against $Ca^{2+}$ [12]. Furthermore, recent findings propose that $Ca^{2+}$ could also be transported by vesicles to the site of calcification [13].

Among the proteins known to mediate $Ca^{2+}$ transport are transient receptor potential (TRP) channels, which are also directly involved in a wide range of functions such as thermal tolerance, heat stress, mechanosensing and $Ca^{2+}$ homeostasis in many organisms. Orthologs of genes that encode TRP channels have been identified in different scleractinian coral species. Most of these relate to physicochemical sensors of environmental parameters such as temperature [14]. However, TRP channel involvement in calcification in marine organisms and how it functions remains unclear. Recently, several studies have highlighted the role of TRP channels in the physiology of marine invertebrates. For example, mollusks such as *Crassostrea gigas* express up to 66 TRP genes, many responsive to thermal stress [15]. while *Mytilopsis sallei* relies on TRPM7 for calcium signaling during larval settlement [16]. In *Chlamys farreri*, 17 TRP channels show spatiotemporal patterns linked to development and sensory functions [17]. Corals from the *Montipora* and *Acropora* genus exhibit TRP expression rhythms associated with environmental cycles, suggesting roles in calcification and reproduction [18,19]. These findings, obtained through transcriptomic analyses, highlight TRP channels as key modulators of physiological adaptation in marine invertebrates.

TRP channels constitute a large and diverse family of proteins that are conserved in many cell types in vertebrates and invertebrates; they permeate cations, including $Ca^{2+}$ and act as signal transducers by regulation of the intracellular $Ca^{2+}$ concentration [20]. Based on the differences in amino acid sequences, the mammalian TRP channels are divided into six subfamilies, including TRPC (Canonical), TRPM

(Melastatin), TRPV (Vanilloid), TRPA (Ankyrin), TRPML (Mucolipin) and TRPP (Polycystin) [21]. The existence of subfamilies that are not found in mammals, TRPN (NOMPC or no mechanoreceptor potential C-like, only found in fish and invertebrates), and TRPVL (VL = voltage-like) suggest that TRP channels have been diversified throughout metazoan evolutionary history [22,23].

TRP channels function as tetramers, with each subunit comprising six transmembrane α-helices (S1–S6), preceded by a cytosolic N-terminal region that, in the TRPV subfamily, contains six ankyrin repeat domains. Following the transmembrane region is an amphipathic TRP helix, which plays a critical role in channel gating and interaction with modulatory molecules. The ion conduction pore is formed by the fifth and sixth transmembrane helices (S5–S6) and a reentrant pore loop that houses the selectivity filter [24]. TRPV5 and the homologous TRPV6 are the two members of the vanilloid subfamily with the highest known $Ca^{2+}$ selectivity and serve as mediators of transepithelial $Ca^{2+}$ transport in mammals [25–27].

TRPV5 and TRPV6 are characterized by strong inward rectification, a unique property within the TRP superfamily that permits the conduction of $Ca^{2+}$ current at negative potentials [28] and constitutive activity, which is supported by the endogenous cofactor, phosphatidylinositol 4,5-bisphosphate (PIP2) [29,30]. Although a limited number of molecules that modulate TRPV5 and TRPV6 have been identified, the antifungals econazole and miconazole are effective inhibitors of these channels [31–33]. While some models of ion transport underlying calcification in corals have been proposed, the cellular mechanisms are poorly understood, and many transporters still need to be identified. The goals of this study were to clone and characterize the biophysical properties of a protein homologous to the TRPV5 channel in the reef-building coral *Pocillopora damicornis.* This study provides insights into the molecular mechanisms underlying coral calcification.

## Materials and methods

### Identification of TRPV5/6-like sequence and cloning

An open-access database of reef coral transcriptomes (http://comparative.reefgenomics.org) was utilized to identify a sequence we recognized as a putative TRPV5/6 channel. The sequence was classified in the National Center for Biotechnology Information (NCBI) as Transient Receptor Potential cation channel subfamily V member 6-like with the accession number XP_027049851.1 from the species *Pocillopora damicornis*. Total RNA was extracted from a small *Pocillopora damicornis* coral fragment acquired from a local aquarium (Reef Services). The method of Chomczynski and Sacchi [34] was used for the extraction of total RNA by incubating a coral sample of approximately 3 cm in 5 ml of solution D (guanidine thiocyanate 4 M, sodium citrate 25 mM, sarcosyl 5% and 2-mercaptoethanol 0.1 M). After incubation, soft tissue was detached from the skeleton with a Pasteur pipette, gently raising and lowering solution D for two minutes. The calcareous skeleton was removed, and the total RNA extraction was continued according to the protocol. Total RNA (1 µg) was obtained, and cDNA was synthesized using SuperScript III Reverse Transcriptase (Invitrogen, USA) and oligo(dT) primers. RT-PCR was performed using the following primers designed against the full open reading frame region of sequence XP_027049851.1.

Forward primer: 5`-TAATTTAGAGGTACCATGGGCAACTGCGCTCAAAAGGCG-3', reverse primer: 5'-GTGATGTTC AGCGGCCGCTCACGTCGATACTGGTGTGAG-3'. The PCR amplification reaction used Phusion DNA Polymerase and the protocol suggested by the manufacturer (Thermo Scientific), with an annealing temperature of 50°C and elongation of 72°C for 1 minute and 35 cycles. Subsequently, additional PCR rounds using new oligos containing restriction sites XbaI-3' and BamHI-5' were performed to amplify the ORF in pJET1.2/blunt and subclone it into pcDNA3.1. All clones and DNA constructs were confirmed by automatic sequencing.

### AlphaFold predicted model

AlphaFold 2.0 was used to predict the structure of a single monomer of the coral TRPV5/6-like channel. The cloned protein sequence was used as input for ColabFold:AlphaFold2 using MMseqs2 notebook [35]. The model was generated with

the default parameters, excluding templates. All predictions were assessed using internally generated confidence scores. Confidence per residue is provided as a predicted Local Distance Difference Test score (pLDDT; scored 0–100) with a pLDDT average of 83.7 (S1 Fig). Structure visualization and the alignment with the rabbit PDB structure (8FFO) [36] were performed using Chimera software [37] by directly comparing the structures to identify similarities.

**Sequence acquisition, alignment and phylogenetic analysis.** TRPV5 and TRPV6 protein sequences from humans were used as a query and blasted against the proteomes and transcriptomes of different selected phyla (Placozoa, Cnidaria, Echinodermata, Porifera, Arthropoda, Mollusca and Chordata) employing the NCBI database (S1 Table). Standard database searches were performed using the non-redundant protein sequences at NCBI. All selected sequences were first analyzed for the presence of six transmembrane domain (TMD) patterns using the DeepTMHMM Server (https://services.healthtech.dtu.dk/service.php?DeepTMHMM). Sequences that did not contain 6 TMDs were discarded.

Only the sequences of transmembrane domains were used for alignment, and the intracellular domains (N- and C-terminal) were identified and deleted due to the presence of too many gaps and mismatches in the alignment. To enable comparison between structure and sequence, transmembrane regions were identified as follows: sequences belonging to Chordata phylum, the TM domains were defined as residues from the start of the S1 domain to the end of the TRP box, as determined by visual inspection of the PDB structure reference (8FFO). In the case of the putative proteins of Placozoa, Cnidaria, Echinodermata, Porifera, Arthropoda and Mollusca, most frequently identified as TRPV5-like and TRPV6-like channels, we performed comparative analyses of the proteins of different organism to determine the TMD based on the AlphaFold predicted structure of the cloned TRPV5/6-like channel and, the AlphaFold Protein Structure Database using the amino acid sequence of predicted models for the TRPV5 or TRPV6 channel of different phyla. The Alphafold models used were: Mytilus coruscus (UniProt Accession Number A0A6J8ACF6), Acromyrmex heyeri (UniProt Accession Number A0A836FK59), Stylophora pistillata (UniProt Accession Number A0A2B4SST2). This analysis allowed us to establish and discard sequences that did not fit the previous parameters.

Amino acid sequences were aligned employing Clustal Omega [38] using default settings, which are incorporated into the JalView 2.11.3.2 software [39] The final alignment contains 177 sequences (S1 Table).

The maximum-likelihood phylogenetic tree was constructed using IQ-tree version 2.3.4 [40], with the Q.pfam+R6 model and 1000 ultrafast bootstrap (UFBoot) [41]. ModelFinder was used to determine the best substitution model [42]. The Interactive Tree of Life (iTOL) v6.9 (unrooted) was used to visualize the phylogenetic tree [43].

To visualize sequence similarity patterns, we performed a sequence alignment using CLUSTAL Omega in Jalview. The program WebLogo 3 [44] was employed to identify and visualize over-represented amino acids within phyla in motif sequences belonging to the TRP box, pore, and econazole binding site, using all 177 sequences.

## Cell culture and heterologous expression

All electrophysiological experiments were performed in HEK293 (ATCC CRL-1573) cells heterologously expressing the human TRPV5 (hTRPV5) and *P. damicornis* TRPV5/6-like (PdTRPV5/6-like) channels. The plasmid containing the human TRPV5 inserted in a pcDNA3.1 vector was donated by Dr. Sharona E. Gordon, University of Washington, Seattle, WA.

HEK 293 cells were grown in 10 cm Petri dishes using complete Dulbecco's Modified Eagle Medium (DMEM, Invitrogen) containing 10% bovine fetal serum (FBS, Invitrogen, USA) and 100 units/ml-100 mg/ml of penicillin-streptomycin (Invitrogen, USA) at 37°C in an incubator with 5% $CO_2$ atmosphere. Cells were subcultured every 4 days by first treating with 0.25% trypsin-ethylenediaminetetraacetic acid (EDTA) for 2 min. Subsequently, 1 ml of DMEM with 10% FBS was added. The cells were mechanically dislodged and reseeded with 2 ml of complete medium in 35 mm culture dishes. Cells at 70% confluence were transiently transfected with different clones prepared with a midiprep kit (Qiagen); 100 ng of plasmid pEGFP-N1 (BD Biosciences Clontech, USA) was cotransfected with cDNA for the different channels to allow transfected-cell identification via their green fluorescence. JetPEI transfection reagent (Polyplus Transfection, France) was

used to transfect the clones according to manufacturer protocol. Electrophysiological recordings were performed one day after transfection.

## Electrophysiology

Current-voltage (I-V) curves were recorded in the outside-out and whole-cell configurations of the patch-clamp technique from HEK293 expressing hTRPV5 or PdTRPV5/6-like. I-V curves were obtained by applying voltage pulses from −150 to +100 mV for 200 ms from a holding potential of 0 mV. To detect currents in the outside-out configuration, membrane patches were excised into a bath solution containing 140 mM NaCl and 10 mM HEPES (pH 7.4). For $Mg^{2+}$ block experiments, 1 mM $MgCl_2$ was added (pH 7.4), and to assess $Ca^{2+}$ currents in whole-cell and outside-out configurations, the bath was exchanged from a monovalent $Ca^{2+}$-free solution composed of 140 mM NaCl, and 10 mM HEPES to a solution containing 110 mM NMDG, 30 mM CaCl2, and 10 mM HEPES.

The standard internal solution for all the experiments contained 140 mM NaCl, 10 mM HEPES, 15 mM EGTA, and 10 mM ATP-diNa (pH 7.3). ATP-diNa (Adenosine 5′-triphosphate disodium salt hydrate) was added to reduce channel rundown. Time-course experiments were carried out by stepping the voltage from −100 to +50 mV for 200 ms in intervals of 3 s at a holding potential of 0 mV for 5 minutes.

To test the effects of miconazole (Sigma-Aldrich), the compound was dissolved in DMSO as a stock at 25 mg/mL and dilutions were made directly in the external recording solution (140 mM NaCl, 10 mM HEPES, and pH 7.4) to the desired concentrations. The highest miconazole concentration (5 µM) had to be applied with a final concentration of DMSO of 2%. Higher concentrations of DMSO inhibited PdTRPV5/6-like currents; thus, we could not apply greater concentrations of miconazole.

To examine the response of PdTRPV5/6-like currents to (-)-menthol (Sigma-Aldrich), we used a bath solution containing 140 mM NaCl, 10 mM HEPES, and pH 7.3. (-)-Menthol was initially dissolved in absolute ethanol as a stock at 50 mM. The highest ethanol concentration was 1% (500 µM (-)-menthol) and did not affect PdTRPV5/6-like currents. All of the tested (-)-menthol concentrations were attained by diluting the stock in the external solution.

For these experiments, the currents were elicited in response to voltage pulses from −100 mV to +50 mV for 200 ms in intervals of 3 s from a holding potential of 0 mV. The response of miconazole and (−)-menthol in PdTRPV5/6-like and hTRPV5-induced currents was performed by applying a different compound concentration to outside-out excised membrane patches from the lowest concentration to the highest. A solution changer with a gravity-driven perfusion system was used to perfuse the compounds during patch-clamp recordings. Each dilution was delivered through a separate tube, so the solutions were not mixed. All chemicals for solutions were acquired from Sigma-Aldrich (Mexico). Pipettes for recording were pulled from borosilicate glass capillaries (Sutter Instrument, USA) and fire-polished to a resistance of 3–5 MΩ. Currents were low-pass-filtered at 2.5 kHz and sampled at 20 kHz with an EPC-10 amplifier (HEKA Elektronik, Germany) and were plotted and analyzed with IGOR Pro (WaveMetrics, Inc.).

## Data analysis

Currents from I-V recordings were normalized to the initial divalent-free solution current at −150 mV and plotted as $I/I_{max}$. For rundown recordings, currents were measured at −100 mV and normalized to the initial current.

Dose–response curves for inhibition of (-)-menthol and miconazole were normalized to the initial current in the absence of the compounds. The dose-response curves were fitted to the Hill equation. All currents were measured at −100 mV.

## Results

### Identification of a coral TRPV5/6 ion channel

The presence of TRP ion channel genes has been documented in transcriptomes of several coral species, although their physiological function is still unknown [14]. Given that TRPV channels are highly permeable to $Ca^{2+}$, we decided to

                                                                       

investigate their presence in coral genome and transcriptome databases. We first identified sequences with homology to TRPV5 and TRPV6 mammalian channels and focused on a sequence from the Pacific coral *Pocillopora damicornis,* which is one of the main reef-building corals in that geographical region.

Employing established procedures, we cloned a full-length transcript encoding the putative TRPV5/6 ion channel. Like other TRP family proteins from vertebrates, this sequence has a predicted protein length of 724 amino acids. Initial sequence comparison with orthologues from two other coral species and human TRPV5 and TRPV6 channels suggests poor conservation from cnidarians to mammals, although several amino acids are absolutely conserved, especially in transmembrane domains and ankyrin repeats, as highlighted in colors in Fig 1A. Of note, the sequence corresponding to the pore domain has a short region of very high conservation across all taxa (LIAM). The region corresponding to the TRP helix, a region important for gating in TRP channels, also has several highly conserved amino acids (S2 Fig).

Structure prediction using Alpha Fold 2 indicates a typical TRPV channel fold, with 6 membrane-spanning alpha helices, an intracellular amino-terminal with six ankyrin repeat domains (ARD) followed by a linker comprised of short helix-turn-helix structures (HLH) and a β-hairpin, which connects the ARD to the pre-S1. Finally, the intracellular carboxy-terminal region comprises a TRP alpha helix domain parallel to the plasma membrane [45] (Figs 1B and S1). It is worth

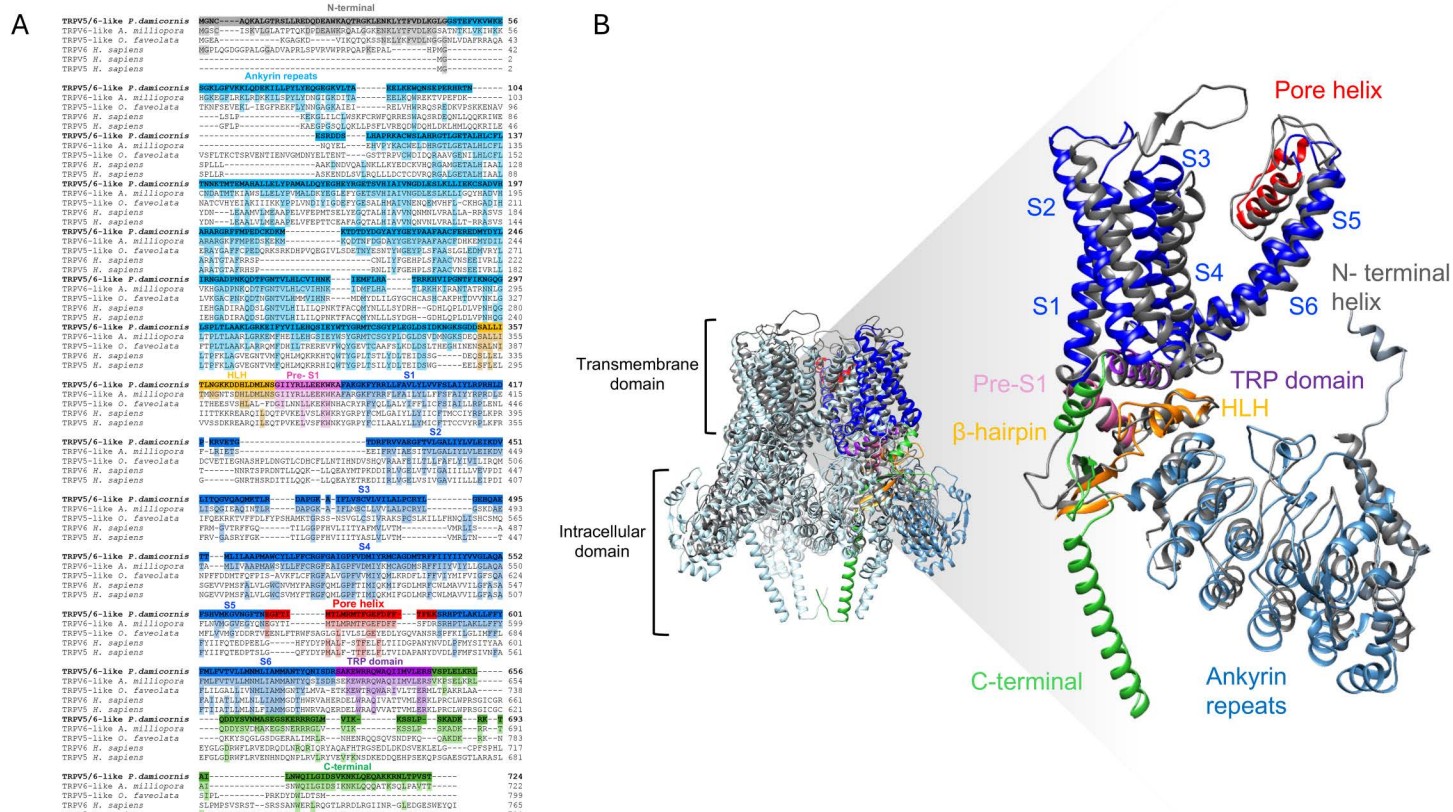

**Fig 1. Amino acid sequence alignment and structural model of *P. damicornis* TRPV5/6-like channel. (A)** Sequence alignment of the PdTRPV5/6-like channel (*P. damicornis*: XP_027049851.1) to stony corals (*A. milliopora* TRPV6-like: XP_029196985.2, *O. faveolata* TRPV5-like: XP_020623877.1), and mammals (*H. sapiens* TRPV5: NP_062815.3, TRPV6: NP_061116.5). Domain organization of PdTRPV5/6-like channels is highlighted in different colors in the sequence alignment and the amino acid conservation, which matches with the PdTRPV5/6-like channel sequence, is shaded. The selected sequences were aligned using ClustalO. **(B)** Left, the PdTRPV5/6-like tetramer model with a single subunit is color-coded and highlighted in the structure. Rabbit (Rb)TRPV5 (PDB:8FFO) is superimposed on the structures and marked in dark gray. Right, side view of the PdTRPV5/6-like monomer predicted by AlphaFold with the functional domains colored to correspond to sequences in **(A)**.

noting that similar structural predictions for related sequences to ours are available in the AlphaFold Protein Structure Database (https://alphafold.ebi.ac.uk/entry/A0A2B4R8J4). Sequence analysis and structural prediction thus suggest that our sequence is related to TRPV5 or TRPV6 ion channels and has a structure that resembles that of the TRPV family.

## Phylogenetic analysis

BLAST searches in genomic and transcriptomic databases identified orthologous proteins to our sequence in other coral species from different clades and homologs from several species from all eukaryotic phyla. Phylogenetic analyses revealed the *P. damicornis* protein sequence is more closely related to so called TRPV5 or 6-like channels from invertebrates than to TRPV5 or TRPV6 mammalian channels. The phylogenetic relationships between the cnidarian TRPV5/6-like channel and other phyla are shown in Fig 2. A maximum likelihood phylogeny inferred from the aligned protein sequences related to TRPV5 and TRPV6 channels from different phyla revealed a deep phylogenetic divergence between Chordates and the Placozoa, Cnidaria, Porifera, Arthropoda, Echinodermata and Mollusca.

   An unrooted phylogenic tree based on protein sequences was generated with the program IQ-tree with the best substitution model Q.pfam+R6. The bootstrap value is given by the color of the branches according to the legend, with the maximum value in green. Scale bar represents the number of amino acid substitutions per site. Different phyla are labeled and shown in distinct background colors. The novel cloned PdTRPV5/6-like channel, grouped in the Cnidaria phylum, is highlighted in red.

   The Chordata TRPV5 and TRPV6 channels are clustered as expected, and all branches have high bootstrap support. In contrast, invertebrate TRPV5 and TRPV6-like sequences formed two groups, each bearing an orthologue from Porifera and Placozoa.

   The *P. damicornis* sequence is clustered with other Cnidarian sequences, including ones from scleractinian species belonging to the genus *Acropora* and *Stylophora*. As mentioned above, some invertebrate sequences found in NCBI, such as *P. damicornis*, are identified as TRPV5-like and TRPV6-like channels. However, it is not clear in the analysis if there is a difference between these groups, as is described for mammalian gene duplication [46]. This analysis also shows that TRPV channel sequences are present in basal organisms like placozoans and suggests cnidarian sequences are ancestral. Finally, our analysis indicates that the TRPV5/6-like channel identified in the present study is a homolog of Chordata TRPV5 or TRPV6 channels, Hence, we refer to our sequence as PdTRPV5/6-like ion channel.

## Electrophysiological properties of PdTRPV5/6-like channel

Next, we investigated whether the PdTRPV5/6-like clone produces ionic currents consistent with TRPV5/6-like channel activity and compared its functional properties to those of human TRPV5 (hTRPV5), one of the most extensively characterized members of this subfamily. Both channels were overexpressed in HEK293 cells, and currents were recorded in whole-cell and outside-out configurations. HEK293 cells do not endogenously express TRPV5 [47], making them a suitable system for heterologous expression of hTRPV5 and PdTRPV5/6-like (S3 Fig).

   In the presence of symmetric $Na^+$, the I-V relationship for both channels shows marked inward currents at negative potentials and almost no outward currents at potentials more positive than 0 mV, indicating monovalent ion conduction and intrinsic inward rectification in the absence of internal or external divalent ions (Fig 3), which has been shown to be a defining characteristic for hTRPV5 and hTRPV6 [27]. $Mg^{2+}$ has been described as a blocker of TRPV5 and TRPV6 channels. In the presence of 1 mM extracellular $Mg^{2+}$, $Na^+$ currents from hTRPV5 are reduced 79.5±9.4% (Fig 3), as previously described for the TRPV5 mammalian channel [48,49]. In contrast, PdTRPV5/6-like currents were reduced only by 61.5±6.3%, indicating that it's less sensitive to blockade by extracellular $Mg^{2+}$.

   Mammalian TRPV5 and 6 channels are specialized in $Ca^{2+}$ transport, at least in some epithelia of vertebrates [50]. Is the PdTRPV5/6-like also a calcium channel? In mammalian TRPV5, $Ca^{2+}$ acts as a permeant blocker at low extracellular concentrations [27]. We decided to compare the $Ca^{2+}$ vs. $Na^+$ permeability of hTRPV5 and PdTRPV5/6-like by assessing

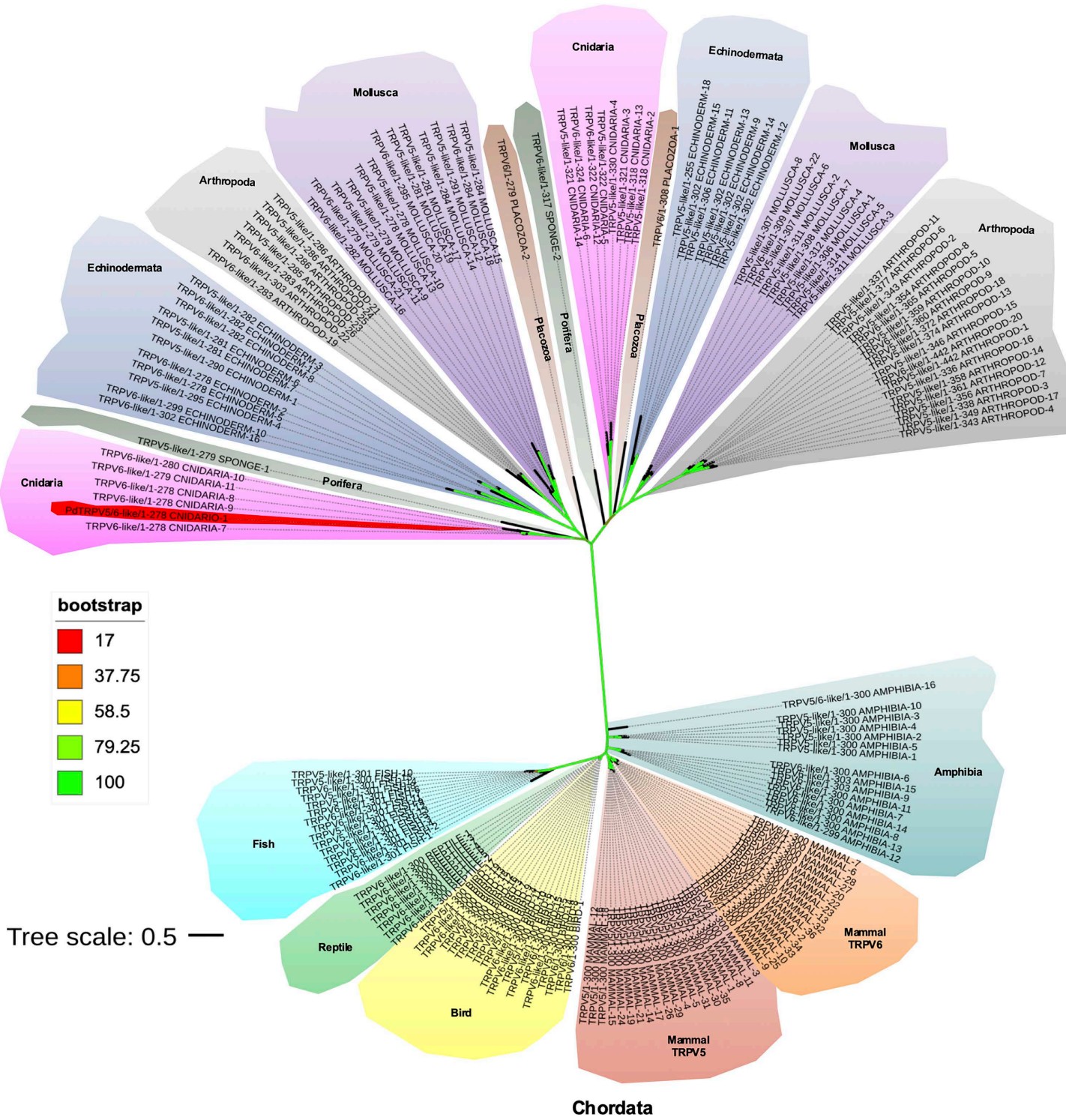

**Fig 2. Phylogenetic tree of TRPV5, TRPV6, and ancestor-like channels.**

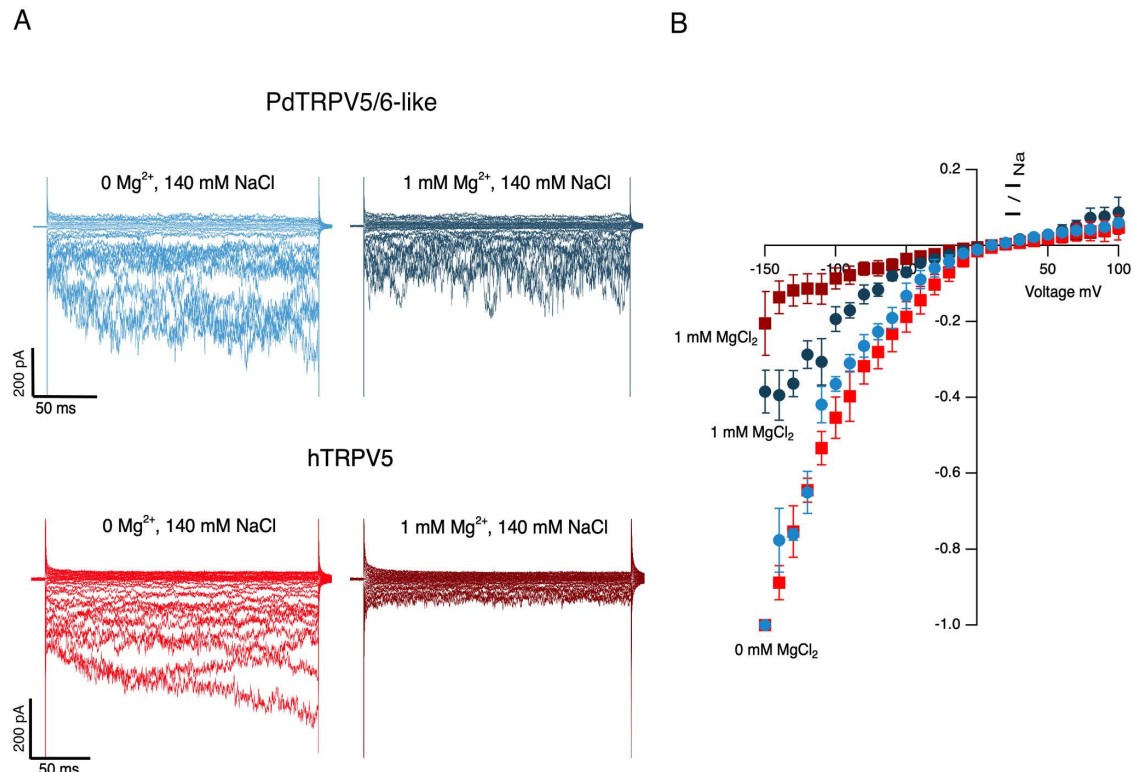

**Fig 3. Representative inward currents of PdTRPV5/6-like vs TRPV5 mammalian channels. (A)** Current traces recorded from HEK293 cells expressing PdTRPV5/6-like channel or hTRPV5 channels in the absence (left) and presence of 1 mM extracellular $Mg^{2+}$ (right) in the outside-out configuration. **(B)** Circles represent I-V curves for PdTRPV5/6-like channels and squares for hTRPV5 channels with divalent cation-free solution (light blue and red) or 1 mM extracellular $Mg^{2+}$ (dark blue and red). Data are represented as mean ± s.e.m. I-V curves were normalized to the maximal current obtained at −150 mV in the absence of divalent cations for each individual experiment (n = 5).

Na+-only and Ca2+-only currents in the same patch. Fig 4 shows currents in the presence of $Na^+$ or $Ca^{2+}$ in the same patch. Interestingly, a comparison of the pore region sequences of the human and rabbit TRPV5 channels with that of the PdTRPV5/6-like channel (Fig 4A) shows that the coral channel has more negatively charged residues in the selectivity filter, which could account for higher $Ca^{2+}$ permeability. For both channels, $Ca^{2+}$ currents are smaller than $Na^+$ currents; however, while both channels conduct inward currents when $Ca^{2+}$ is the only permeant cation, the fraction of $Ca^{2+}$ to $Na^+$ current in the coral channel is significantly higher (27.4 ± 2.9%) than for the mammalian channel (11.1 ± 1.5%) (Fig 4D).

**Miconazole inhibits hTRPV5, but not PdTRPV5/6**

Miconazole and econazole are antifungals with similar structures identified as effective blockers of mammalian TRPV5 and TRPV6 channels [33]. The econazole-binding pocket in rabbit TRPV5 was identified by cryo-EM between the S3–S4 helices, the S4–S5 linker, and the S6 helix of the adjacent subunit [31]. An equivalent site exists in the mammalian TRPV6 channel [32]. To determine miconazole inhibition, we compared the effect of the blocker on hTRPV5 and PdTRPV5/6-like currents.

In the absence of modulators, TRPV5 is a constitutively active channel [27], but needs PIP2 to remain active [51]. In the absence of PIP2, such as during whole-cell or cell-free recording configurations, the current tends to rundown in response to the repetitive stimulation [27]. To reduce rundown, we evaluated the application of intracellular ATP-Na in the

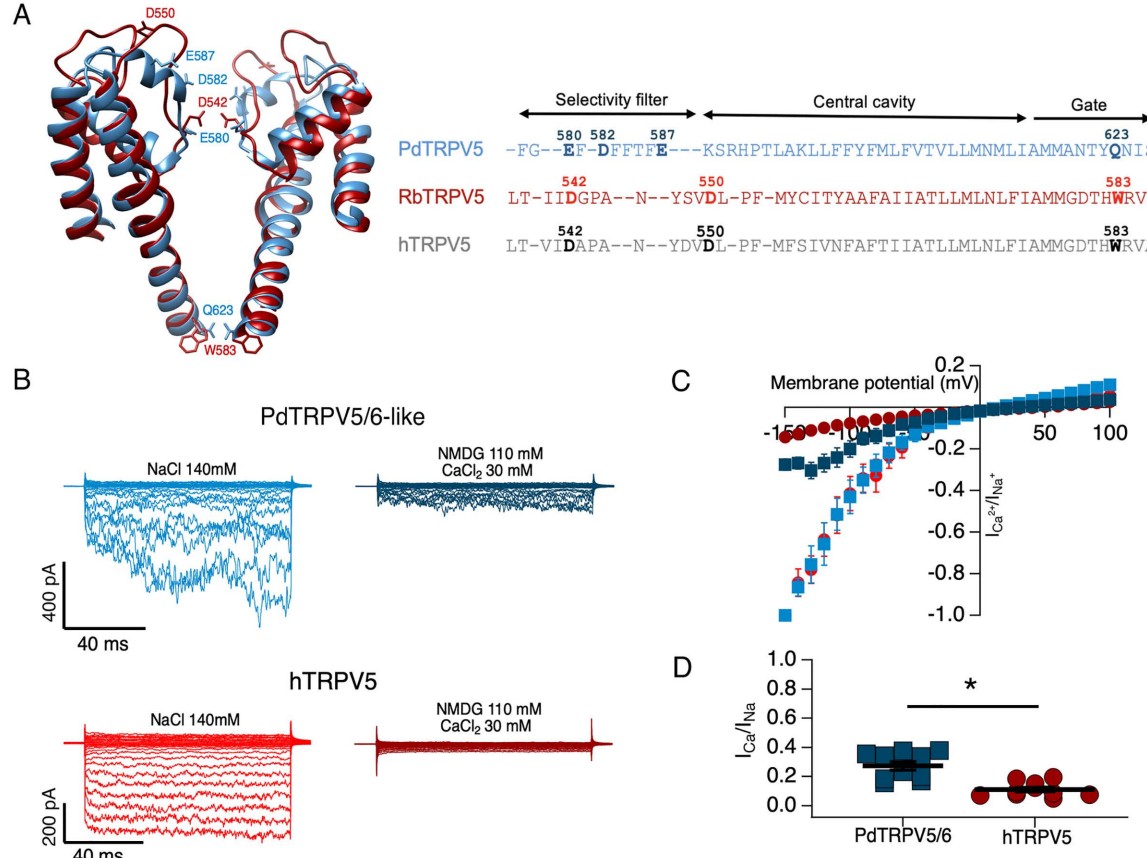

**Fig 4. Pore structure comparison and permeation of Ca²⁺ through PdTRPV5/6-like and hTRPV5 channels. (A)** Left panel, comparison of the predicted PdTRPV5/6-like channel pore domain structure (blue) with rabbit RbTRPV5 (PDB 8FFO) (red). (Right) The sequence alignment shows negatively charged residues proposed to be important for calcium selectivity in mammalian channels (bold letters). **(B)** Current traces recorded from HEK293 cells expressing PdTRPV5/6-like or hTRPV5 channels exposed to divalent cation-free solution (left panels) or 30 mM extracellular Ca²⁺ and with Na⁺ replaced by NMDG⁺ (right panels) in the outside-out configuration. **(C)** I-V curves of currents carried by Na⁺ or carried by 30 mM Ca²⁺ in the absence of another permeable cation in PdTRPV5/6-like (blue squares) and hTRPV5 channels (red circles), respectively. **(D)** Average fraction of Ca²⁺ current obtained from data in experiments in **(C)**. Ca²⁺ currents were normalized to the initial Na⁺ current obtained at −150 mV without divalent cations. Data are mean ± s.e.m. of 10 experiments. Statistical comparison was made using Student's *t-test* *P < 0.05.

absence of divalent cations, as previously described as means to stabilize the interaction between PIP₂ and the channel [48,52]. We found that in these conditions, inward current progressively remained stable and even tended to increase (runup) during the 5 minutes of the experiment. This behavior is also present in the coral TRPV5/6-like channel (S4 Fig). Miconazole inhibition of hTRPV5 and PdTRPV5/6-like channels was compared using outside-out membrane patches in the presence of intracellular ATP-Na to avoid rundown during blocker application. Extracellular application of miconazole produced inhibition of hTRPV5 channels, as reflected by reduced current amplitudes (Fig 5A), and we determined that the half-inhibition concentration (IC₅₀) was 51 nM (Fig 5B).

In contrast, miconazole did not produce a significant current decrease in the PdTRPV5/6-like channel, even at high concentrations (5 µM). Comparing the 177 sequences in the region that forms the miconazole/econazole binding site shows that, out of the three residues important for drug binding in mammalian channels, only phenylalanine is present in the coral PdTRPV5/6-like channel, consistent with the absence of miconazole block (Fig 5C). Furthermore, the site is present in TRPV5 channels from vertebrates but is absent in sequences from invertebrates (Fig 5D).

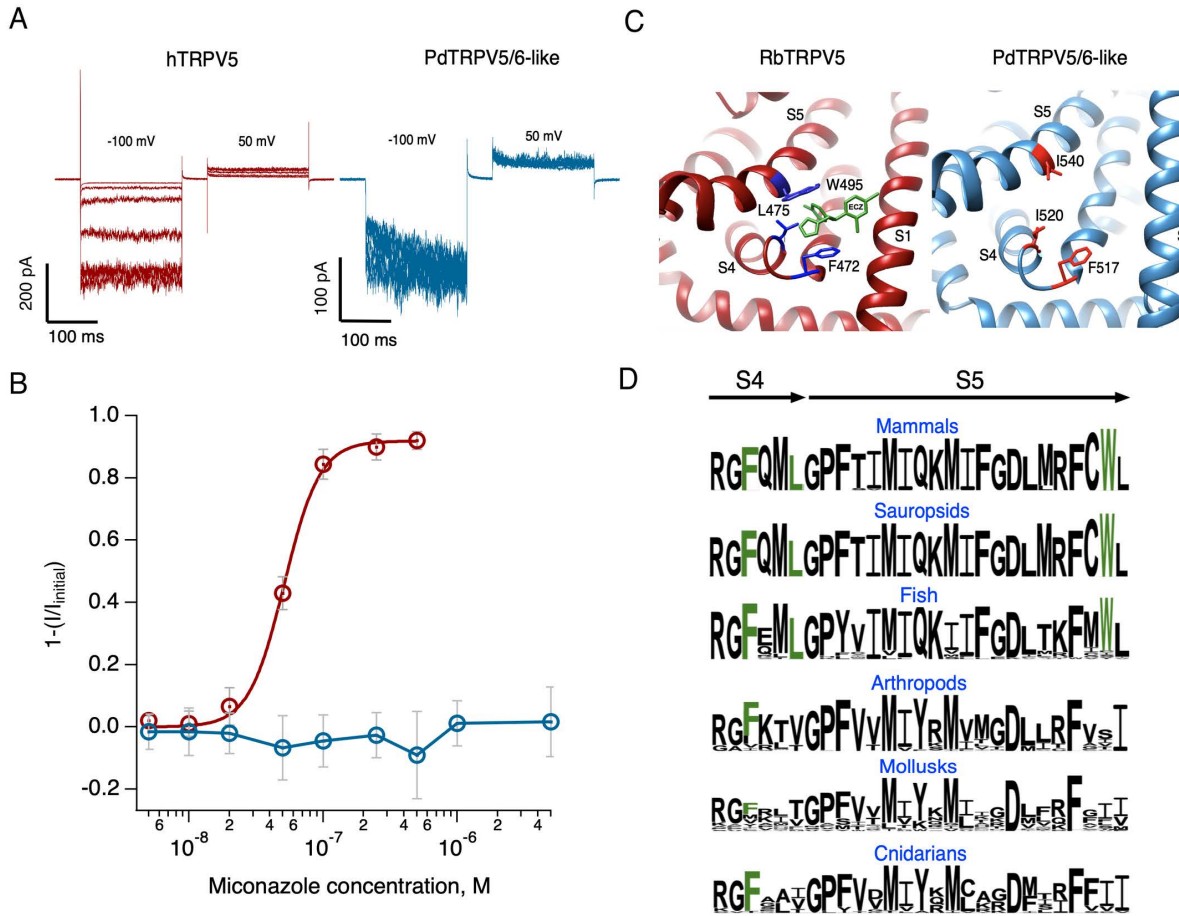

**Fig 5. Inhibition of hTRPV5 and PdTRPV5/6-like channels by miconazole. (A)** Representative currents at indicated miconazole concentration points in (B) recorded at −100 mV in outside-out configuration. **B)** Fraction of blocked or inhibited current. Currents at −100 mV were normalized to the corresponding current in the absence of the blocker. The smooth red curve is a fit to the Hill equation ($IC_{50}$ = 51.2 nM and Hill coefficient = 3.2) of the hTRPV5 channel data. Blue circles show no effect of miconazole on the PdTRPV5/6-like channel. All concentrations were tested in each outside-out membrane patch (n = 6). Data are mean ± s.e.m. **(C)** Left, structure of the RbTRPV5 (8TF3) econazole binding site. Right, predicted structure of the corresponding region in PdTRPV5/6-like. Residues important for interaction with econazole and their corresponding residues are highlighted as sticks. **(D)** Consensus amino acid sequences from the indicated phyla, with residues interacting with econazole highlighted in green.

## Menthol-induced inhibition of the PdTRPV5-like channel

It has been shown that menthol, a plant-derived monoterpene, can be used to produce symbiont expulsion, leading to coral bleaching [53]. Since menthol can act as an inhibitor or activator of several TRP channels from vertebrates [54,55] and TRP channels may play a role in coral bleaching [56], we decided to explore if menthol has effects on the PdTRPV5/6-like channel. To assess the possible actions of menthol, we applied it to outside-out patches expressing the PdTRPV5/6-like channel in DVF solutions.

As illustrated in Fig 6, the PdTRPV5/6-like channel exhibited current inhibition in response to menthol in a concentration-dependent manner. In our patch-clamp recordings, (-)-menthol decreases the PdTRPV5/6- like currents with an $IC_{50}$ of 120.46 μM (Hill coefficient: 0.933, n = 8). This responsiveness range appears to be higher and opposite than the effect of (-)-menthol in its selective activation of another TRP channel, TRPM8, with a reported $IC_{50}$ of 62 μM in the whole-cell configuration [57].

## Discussion

The role of ion channels in coral physiology has just begun to be functionally studied. Not many ion channels have been identified or studied and this is an important emerging area of research [14,58,59]. In this work, we present the cloning and characterization of a TRPV channel from the Pacific coral *Pocillopora damicornis*. The amino acid sequence shows homology with TRPV5 and TRPV6 channels from mammals and other vertebrates, albeit with little overall sequence conservation. However, similar sequences are present in several invertebrates, including other cnidarians, sponges, and even a placozoan, indicating that these sequences might be ancestral to the TRPV5/TRPV6 ion channels found in vertebrates. It has been shown that the distinction between TRPV5 and TRPV6 as separate genes is present in mammals, sauropsids, and amphibians, but not in lower vertebrates such as birds, cyclostomes, bony fish, and coelacanths [46,60].

We think that invertebrate sequences cannot be easily identified as either TRPV5 or TRPV6, and thus, we propose that they be referred to as TRPV5/6-like. Here, we expand on phylogenetic comparisons of these channels to include our coral sequence and find that it belongs to an ancestral group of channels from which later, through gene duplication in vertebrates, TRPV5 and TRPV6 arose.

However low the sequence similarity between invertebrate and vertebrate channels is, sequence analysis shows that there is conservation in regions that are important for the distinct function of TRPV5 and TRPV6 channels, such as the pore domain, important for ion permeation, and the TRP box, which determines several gating characteristics in TRP channels (S2 Fig) [61,62]. Moreover, the AlphaFold-predicted structure of the TRPV5/6-like channel monomer closely resembles other members of the TRPV family. Despite low sequence conservation, the model demonstrates high structural reliability, with an average pLDDT score of 83.4 and values exceeding 90 in the intracellular regions. In contrast, lower scores (<70), primarily observed at the C-terminus, likely correspond to intrinsically disordered segments (S1 Fig).

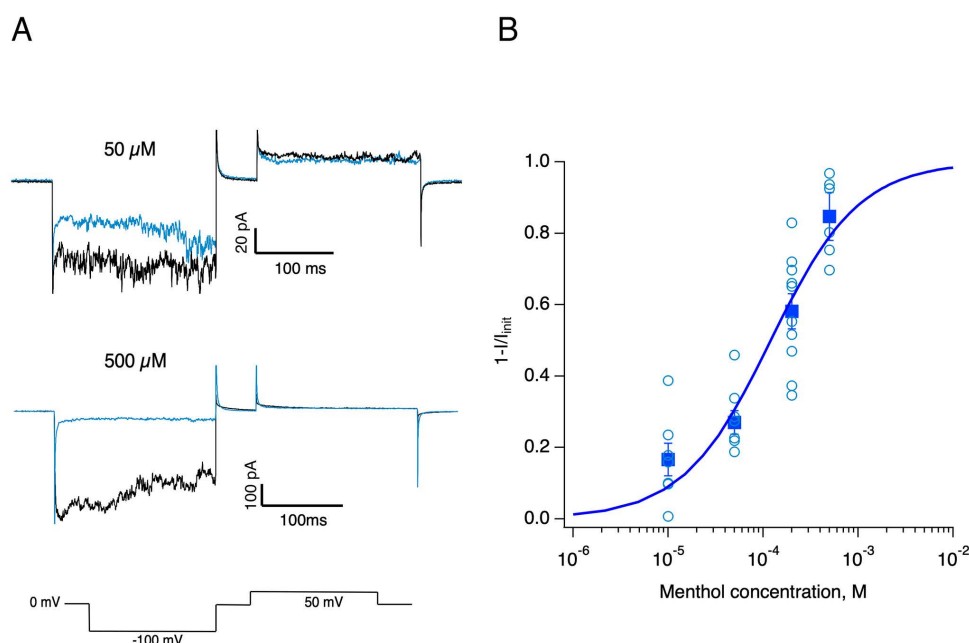

**Fig 6. Menthol inhibition of the PdTRPV5/6-like channel. A)** Effect of menthol on Na+ current amplitude through PdTRPV5/6-like channels in outside-out patches. Representative control current traces (black) and after application of the indicated menthol concentration (blue) trace. The bottom lines illustrate the voltage protocol. **B)** Dose-response curve of menthol inhibition. Empty circles are fractional inhibition of Na+ currents at the indicated concentration measured at −100 mV. Data from n = 10 experiments. Squares indicate the mean, and the error bars are the s.e.m. IC$_{50}$ value of 120.46 µM was obtained from a fit to the standard Hill equation with a Hill coefficient of 0.93.

Even though the overall structure is conserved, key functional differences exist between vertebrate and invertebrate channels. For example, in contrast to vertebrate TRPV5 and TRPV6, the coral channel is not blocked by miconazole, possibly because it lacks a complete binding site. A comparison of consensus sequences from several phyla, shows an evolutionary succession leading to the vertebrate channel's miconazole binding site, which is incomplete in invertebrate sequences, but appears in some fish sequences (Fig 5D), probably by acquisition of the rest of the important residues.

Other important differences are related to the block and permeation of divalent cations. Although the coral TRPV5/6-like channel behaves similarly to mammalian TRPV5 or TRPV6 channels in that it is a constitutively active inward rectifier, we show that the coral channel is more permeable to $Ca^{2+}$ than the human homolog and, it also seems more resistant to extracellular $Mg^{2+}$ block. The degree of conservation of the pore sequence is low, but several features are preserved. For example, the rabbit TRPV5 channel has two negatively charged residues, D542 and D550, which are also present in humans and have been shown to contribute to $Ca^{2+}$ permeability [62]. In contrast, the PdTRPV5/6-like has an extra residue with a negative charge, D582, which may account for increased permeability to $Ca^{2+}$ in this channel (Fig 4A).

The higher $Ca^{2+}$ permeability and lower sensitivity to $Mg^{2+}$, might reflect an important functional role of this channel in coral physiology, allowing more efficient $Ca^{2+}$ entry that could regulate calcification and a smaller resting fraction of blocked channels due to higher $Mg^{2+}$ concentration in seawater. It has been suggested that voltage-gated $Ca^{2+}$ channels (VGCC) play a role in transcellular $Ca^{2+}$ transport needed for calcification [11]; however, VGCC needs large depolarizations to open, limiting their usefulness in vectorial $Ca^{2+}$ transport. In contrast, the TRPV5/6-like channel is constitutively open and would offer an open pathway at the resting potential of coral cells (which is unknown), allowing for efficient calcium entry into the calcifying cells. We propose that coral TRPV5/6-like channels are responsible for $Ca^{2+}$ entry into calcifying cells, a role reminiscent of the function of TRPV5 or 6 in other $Ca^{2+}$ absorptive epithelia.

One intriguing finding in our work is the fact that the monoterpene menthol can produce inhibition of the PdTRPV5/6 channel. This is relevant because menthol is commonly used to experimentally induce rapid coral bleaching [53]. Coral bleaching is triggered by environmental stressors often linked to climate change, resulting in the loss of symbiotic dinoflagellates and compromising the coral's energy supply, contributing to reef degradation. The molecular mechanism by which menthol acts as an effective experimental bleaching agent inducing the loss of symbionts remains unknown. It has been proposed that menthol, at mM concentrations, inhibits photosystem II (PSII) activity in *Symbiodinium* symbionts residing within coral tissues, ultimately leading to their expulsion [63]. Furthermore, it is known that some terpenes including menthol, can induce cellular stress responses in plants by disrupting membrane integrity, including those of chloroplasts [64], causing rapid membrane depolarization similar to cold stress [65], suggesting that menthol acts as a cellular stressor across diverse biological systems. Interestingly, transcriptomic analyses of corals under natural bleaching conditions (elevated temperature and light) have revealed downregulation of genes involved in calcium signaling, including members of the TRP channel family [56]. The observation here that menthol can block TRPV5/6 coral channels suggests an intrinsic mechanism involving the dysregulation of calcium homeostasis. Menthol might block the entry of $Ca^{2+}$ into the calcifying cells via PdTRPV5/6-like channels and induce cellular stress, which could lead to symbiont expulsion and coral bleaching. Interestingly, the concentrations of menthol employed to induce bleaching are very similar to those that block TRPV5/6-like channels. This hypothesis will have to be tested in isolated coral cells.

Our work is a steppingstone to understanding the role of ion channels in coral physiology, particularly, the molecular basis of calcification. However, several future questions need to be answered, such as protein localization and its in situ function in calcifying cells.

## Conclusions

In this work we have identified and characterized a novel ion channel from the coral *Pocillopora damicornis* that has characteristics of TRPV5 and TRPV6 ion channels from vertebrates. This TRPV5/6-like ion channel has permeability to sodium and calcium ions, but, unlike its vertebrate homologs, is not blocked by miconazole, a well-characterized inhibitor

of TRV5 and 6 channels. Interestingly, we show that the coral channel is blocked by menthol, a terpene that has been used to induce coral bleaching in experimental settings. Our findings open the door to an increased understanding of the role of ion channels in coral physiology.

## Supporting information

**S1 Table. Accession numbers of TRPV5 and TRPV6 sequences used in this study.** List of TRPV5 and TRPV6 protein sequences retrieved from NCBI grouped by phylum. The table includes the species name and corresponding accession number for each sequence.
(DOCX)

**S1 Fig. Confidence scores of AlphaFold-generated model of the PdTRPV5/6-like channel.** Alphafold model colored by the predicted local distance difference test score (pLDDT). A pLDDT ≥ 90 have very high model confidence, residues with 90 > pLDDT ≥ 70 are classified as confident, while residues with 70 > pLDDT > 50 have low confidence. Overall structure scores are provided in the table. Interface pTM scores provide the confidence score for the complete model (scored 0–1).
(TIF)

**S2 Fig. Sequence logo visualization for the pore and TRP box in TRPV5, TRPV6 and ancestor-like channels.** (A) Consensus amino acid sequences of the pore domain (selectivity filter, central cavity, and gate) for the indicated groups of organisms. The colored residues are the most conserved or demonstrated to have important functional roles in channel behavior. (B) Consensus amino acid sequences for the TRP box domain depicting highly conserved residues across mammals, sauropsids, fish, arthropods, mollusks, and cnidarians.
(TIF)

**S3 Fig. HEK293 cells lack endogenous TRPV5 channel expression.** (A) Representative whole-cell current traces from untransfected HEK293 cells and cells transfected with either human TRPV5 or coral PdTRPV5/6-like channels. (B) The corresponding whole-cell current–voltage (I–V) relationships are shown, with error bars representing the standard error of the mean (± s.e.m.).
(TIF)

**S4 Fig. Effect of intracellular ATP-diNa in PdTRPV5/6-like and hTRPV5 channels.** (A and C) Time course (5-min with 3s intervals) of monovalent currents recorded at −100 mV after patch excision in the outside-out configuration (light blue and orange circles) in the absence of ATP. The inward currents increased over time in the presence of 10mM ATP-diNa (n = 5) (dark blue and red circles) for both species, respectively. (B and D) Representative current traces with prolonged exposure to ATP-diNa, which prevents rundown of TRPV5 channels in symmetrical conditions of monovalent cations. The dotted line indicates the final current after 5 minutes. Normalized currents show the average of each experiment with error bars for both conditions (shadows). Data are mean ± s.e.m.
(TIF)

## Acknowledgments

K. A. Méndez-Reséndiz is a doctoral student from the Programa de Doctorado en Ciencias Biomédicas, Universidad Nacional Autónoma de México (UNAM) and received a fellowship from the Consejo Nacional de Humanidades, Ciencia y Tecnología (CONAHCyT; 781803).

This work is in fulfillment of the requirements for a doctoral degree of the Programa de Doctorado en Ciencias Biomédicas for K. A. Méndez-Reséndiz at the Universidad Nacional Autónoma de México. We thank Itzel Llorente for her technical

assistance with cell culture and transfection, and Laura Ongay from the Molecular Biology Facility at the Instituto de Fisiología Celular, UNAM, for DNA sequencing. We also acknowledge Augusto César Poot Hernández from the Bioinformatics and Information Management Unit at the same institute for his support with phylogenetic tree analyses, and Elsa Evaristo for her valuable technical support.

## Author contributions

**Conceptualization:** Angélica Méndez-Reséndiz, Tamara Rosenbaum, León D. Islas.

**Data curation:** Angélica Méndez-Reséndiz.

**Formal analysis:** Angélica Méndez-Reséndiz, León D. Islas.

**Funding acquisition:** Tamara Rosenbaum, León D. Islas.

**Investigation:** Angélica Méndez-Reséndiz, Gisela E. Rangel-Yescas, Miguel Benítez-Angeles.

**Methodology:** Angélica Méndez-Reséndiz, Miguel Benítez-Angeles, León D. Islas.

**Project administration:** Angélica Méndez-Reséndiz, León D. Islas.

**Resources:** Gisela E. Rangel-Yescas, León D. Islas.

**Supervision:** Tamara Rosenbaum, León D. Islas.

**Validation:** Tamara Rosenbaum.

**Visualization:** Angélica Méndez-Reséndiz, León D. Islas.

**Writing – original draft:** Angélica Méndez-Reséndiz, Tamara Rosenbaum, León D. Islas.

**Writing – review & editing:** Angélica Méndez-Reséndiz, Gisela E. Rangel-Yescas, Miguel Benítez-Angeles, Tamara Rosenbaum, León D. Islas.

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
