## [Decision Letter · Decision Letter 0]

13 Dec 2024

Dear Dr. Islas Suarez,

Thank you for submitting your manuscript to PLOS ONE. After careful consideration, we feel that it has merit but does not fully meet PLOS ONE’s publication criteria as it currently stands. Therefore, we invite you to submit a revised version of the manuscript that addresses the points raised during the review process.

**ACADEMIC EDITOR: **

Authors have overexpressed TRPV5/6-like channel in HEK293, but no supporting data were given. In absence of internal protein, role of new TRPV5/6-like channel may be explored to strengthen the concept.In figure 4, legend does not tally with the respective panels. Authors may extend writing in the text for figure 4 mentioning each panel.Various references and its proper citation need to be thoroughly checked.

Indicate which changes you require for acceptance versus which changes you recommendAddress any conflicts between the reviews so that it's clear which advice the authors should followProvide specific feedback from your evaluation of the manuscript

We look forward to receiving your revised manuscript.

Kind regards,

Chandi C. Mandal, Ph.D.

Academic Editor

PLOS ONE

“Dirección General de Asuntos del Personal Académico, Programa de Apoyo a Proyectos de Investigación e Innovación Tecnológica (DGAPA-PAPIIT) Grant No. IN201824. Fronteras de la Ciencia grant No. FC-513 from Consejo Nacional de Humanidades, Ciencia y Tecnología”

“K. A. Méndez-Reséndiz is a doctoral student from the Programa de Doctorado en Ciencias Biomédicas, Universidad Nacional Autónoma de México (UNAM) and received a fellowship from the Consejo Nacional de Humanidades, Ciencia y Tecnología (CONAHCyT; 781803).

This work is in fulfillment of the requirements for a doctoral degree of the Programa de Doctorado en Ciencias Biomédicas for K. A. Méndez-Reséndiz at the Universidad Nacional Autónoma de México. This research was funded by the Dirección General de Asuntos del Personal Académico, Programa de Apoyo a Proyectos de Investigación e Innovación Tecnológica (DGAPA-PAPIIT) Grant No. IN201824 and by a Fronteras de la Ciencia grant No. FC-513 from Consejo Nacional de Humanidades, Ciencia y Tecnología. We thank Laura Ongay from the Molecular Biology Facility of the Instituto de Fisiología Celular at UNAM for DNA sequencing, Augusto Cesar Poot Hernández, Unidad de Bioinformática y Manejo de la Información, Instituto de Fisiología Celular at UNAM for help with phylogenetic trees. We also thank Itzel Alejandra Llorente and Elsa Evaristo for technical support.”

“Dirección General de Asuntos del Personal Académico, Programa de Apoyo a Proyectos de Investigación e Innovación Tecnológica (DGAPA-PAPIIT) Grant No. IN201824. Fronteras de la Ciencia grant No. FC-513 from Consejo Nacional de Humanidades, Ciencia y Tecnología”

4. In this instance it seems there may be acceptable restrictions in place that prevent the public sharing of your minimal data. However, in line with our goal of ensuring long-term data availability to all interested researchers, PLOS’ Data Policy states that authors cannot be the sole named individuals responsible for ensuring data access (http://journals.plos.org/plosone/s/data-availability#loc-acceptable-data-sharing-methods).

Additional Editor Comments:

Manuscript has reviewed by reviewers and editor. Based on their opinions, it may be revised. Authors need to address all comments made by reviewers and editor.

Editor comments: • Authors have overexpressed TRPV5/6-like channel in HEK293, but no supporting data were given. In absence of internal protein, role of new TRPV5/6-like channel may be explored to strengthen the concept.

• In figure 4, legend does not tally with the respective panels. Authors may extend writing in the text for figure 4 mentioning each panel.

• Various references and its proper citation need to be thoroughly checked.

Reviewers' comments:

Reviewer's Responses to Questions

**Comments to the Author**

Reviewer #1: The paper is dedicated to the identification of novel channels of the TRP family in stony corals. The proteins of this family are known to be able to transport Ca2+ ions, and thus may participate in the calcification processes. This is of special interest in the case of corals, in which the mechanisms of skeleton formation are poorly understood.

The paper is relevant to the field, as it identifies at possible molecular component of the skeleton formation. The finding is novel, albeit not unique, as there are other candidates known to transport Ca2+ ions into the cells of the corals, and at least orthologs of the TRP channel genes were identified in the corals. However, the introduction does not provide enough data on whether other TRP channels are known in corals. The literature, however, does not seem to contain many papers on this topic, so the present paper is quite novel in the part of the protein cloning and characterization of its function, although the gene candidate was already known.

The methodology of the paper is reasonable. The authors found the gene candidate, which was already annotated in a database, cloned it, transfected into the HEK 293 cells and used electrophysiology to characterize its responses. Additionally, they used the AlphaFold software to produce a predicted 3D structure of the protein.

The conclusions are supported by the data.

The paper could be published after a minor revision.

Specific points

1. The introduction is lacking data on whether there are other TRP cahnnels identified in corals

2. The validity of the protein 3D structure produced by AlphaFold should be discussed

3. It would be of significant interest to discuss the physiological meaning of the menthol inhibition of the coral TRP channels. Is it a channel feature exploited by plants or is it an overstimulated part of the coral's own physiology?

---

## [Author Response · Author response to Decision Letter 1]

14 Aug 2025

Response to Reviewers

Authors have overexpressed TRPV5/6-like channel in HEK293, but no supporting data were given. In absence of internal protein, role of new TRPV5/6-like channel may be explored to strengthen the concept.

Thank you for your insightful comment. In response to your question, we have attached a new supplementary figure 3, demonstrating that, in the absence of human TRPV5 or coral PdTRPV5/6-like channel transfection in HEK293 cells, no currents are observed. This further supports the absence of endogenous TRPV5 currents in these cells. Additionally, we have included a new reference (line 383, reference 47) confirming that HEK293 cells do not express endogenous TRPV5 currents, which strengthens the validity of our findings.

In figure 4, legend does not tally with the respective panels. Authors may extend writing in the text for figure 4 mentioning each panel.

Thank you for your observation. We have attached a new Figure 4, which matches the descriptions of each panel provided in the figure legend, and corrected the text accordingly (line 409 with Fig.4A)

Various references and its proper citation need to be thoroughly checked.

Thank you for your valuable comment. We have carefully reviewed all references and ensured that they are appropriately cited. Any inconsistencies have been corrected in the revised manuscript to match the references.

The introduction is lacking data on whether there are other TRP cahnnels identified in corals

We have revised the introduction to include additional information on TRP channel expression in marine organisms, with a particular focus on findings from transcriptomic studies. We now highlight the identification of TRP family members in bivalves, which significantly contribute to the available transcriptomic data on expression patterns across marine invertebrates. Regarding corals, we have incorporated all currently available references reporting the presence of multiple TRP channel gene transcripts in two species, Montipora capitata and Acropora digitifera (line 70-80, references 15,16,17,18,19) Additionally, to reinforce this point, we cited a comparative genomic study (line 72, reference 14) that identified orthologues of several TRP family members across different coral species.

The validity of the protein 3D structure produced by AlphaFold should be discussed

The AlphaFold model was assessed using the predicted Local Distance Difference Test (pLDDT) score. The model achieved an average pLDDT of 83.4, which is considered confident, with values exceeding 90 in the majority of well-structured regions, indicating very high confidence in these areas. Lower scores (<70) were mainly observed at the C-terminal end. We have now included a statement in the discussion (line 521) referencing these values and Supplementary Figure 1, which specifically illustrates the regions with the greatest score variability. Additionally, we support the reliability of our structural model by referencing other TRPV monomer coral structures available in the AlphaFold Protein Structure Database in the Results section (line 217). Altogether, these elements reinforce the structural validity of our model.

It would be of significant interest to discuss the physiological meaning of the menthol inhibition of the coral TRP channels. Is it a channel feature exploited by plants or is it an overstimulated part of the coral's own physiology?

Thank you for your valuable comment. We greatly appreciate it, as it has helped us enhance our discussion on the effects of menthol on the coral channel. The use of menthol as a bleaching agent has been widely adopted in laboratory settings to experimentally induce bleaching under controlled conditions. For this reason, the effects of menthol on TRP channels represent a novel and emerging area of research. We have expanded the discussion to clarify the context in which menthol is relevant to exploring the bleaching phenomenon in corals.

The only available mechanistic reference suggests that menthol inhibits photosystem II (PSII) in symbionts. To expand the response, we have included additional examples of menthol's cellular effects in other systems, such as plants, where it disrupts cellular membranes and alters membrane potential (line 562, reference 64,65). While studies on menthol in these systems are limited, we support the idea that menthol acts as a stressor agent. However, there is no evidence to suggest that TRP channels are involved in these effects in plants. In this context, we have included a reference that discusses changes in gene expression patterns associated with calcium signaling in response to environmental bleaching conditions, further reinforcing the potential link between bleaching and TRP channels (line 567, reference 56).

To our knowledge, our study is the first to report a functional role for a native TRPV5/6-like channel in coral physiology, specifically in the context of bleaching induced by menthol. We suggest that menthol disrupts calcium homeostasis in coral cells by inhibiting the TRPV5/6-like channel, leading to changes in calcium levels. This disruption could contribute to the initiation of the bleaching response, suggesting that the process may be, at least in part, mediated by the coral's own physiological mechanisms, rather than solely by damage to the algal symbionts.

---

## [Editor Report · Decision Letter 1]

3 Sep 2025

A novel TRPV5/6-like channel from a scleractinian coral that is inhibited by menthol.

PONE-D-24-30962R1

Dear Dr. Islas Suarez,

We’re pleased to inform you that your manuscript has been judged scientifically suitable for publication and will be formally accepted for publication once it meets all outstanding technical requirements.

Kind regards,

Chandi C. Mandal, Ph.D.

Academic Editor

PLOS ONE

Additional Editor Comments (optional):

The revision has significantly improved quality of the manuscript. 
---

## [Editor Report · Acceptance letter]

PONE-D-24-30962R1

PLOS ONE

Dear Dr. Islas,

I'm pleased to inform you that your manuscript has been deemed suitable for publication in PLOS ONE. Congratulations! Your manuscript is now being handed over to our production team.

Kind regards,

on behalf of

Dr. Chandi C. Mandal

Academic Editor

PLOS ONE